# The Variation of Serotonin Values in Dogs in Different Environmental Conditions

**DOI:** 10.3390/vetsci9100523

**Published:** 2022-09-25

**Authors:** Timea Andrea Bochiș, Kálmán Imre, Simona Marc, Cristina Vaduva, Tiana Florea, János Dégi, Octavian Sorin Voia, Călin Pop, Ioan Ţibru

**Affiliations:** 1Faculty of Veterinary Medicine, Banat’s University of Agricultural Sciences and Veterinary Medicine “King Michael I of Romania”, 300645 Timișoara, Romania; 2Faculty of Animal Resources Bioengineering, Banat’s University of Agricultural Sciences and Veterinary Medicine “King Michael I of Romania”, 300645 Timișoara, Romania

**Keywords:** serotonin level, emotional states, canine

## Abstract

**Simple Summary:**

The multiple implications of serotonin in behavior manifestations have shaped the goal of the present study, which was to evaluate the variation of serum serotonin levels in different experimental groups of dogs to establish whether serum serotonin levels could serve as indicators of aggressive behavior, especially when adoption is considered. The experimental groups were divided into three variants: Variant 1—two groups of medium (*n* = 6) and small (*n* = 4) breed shelter dogs; Variant 2—dogs with owners (*n* = 15) and dogs without owners but in foster care (*n* = 10), after administration of pre-spaying/neutering anesthesia; and Variant 3—dogs in different behavioral states (*n* = 8), classified as follows: M1—happy, M2—aggressive, M3—calmed status, post-exposure to a stressful situation, compared to the reference time referred to as M0. Significant results were found between M1 and M2 (*p* ≤ 0.05, decrease of serotonin by 89.61 ng/mL), as well as between M2 and M3 (*p* ≤ 0.008, increase by 112.78 ng/mL). Following anesthesia, the average mean serotonin values were significantly lower (*p* ≤ 0.003), by 63.85 ng/mL, in stray dogs compared to dogs with owners, leading to a presumptive conclusion that serotonin levels could serve as indicators for potentially aggressive behaviors.

**Abstract:**

Serotonin is considered to be the neurotransmitter that controls several types of behavior: aggressiveness, impulsivity, food selection, stimulation, sexual behavior, reaction to pain, and emotional manifestations. The aim of this study was to determine the serotonin values in 43 dogs, divided into three different experimental variants: (1) between two groups of medium (*n* = 6) and small (*n* = 4) breed shelter dogs; (2) in dogs with (*n* = 15) and without (*n* = 10) owners after administration of pre-spaying/neutering anesthesia; (3) in different behavioral states (*n* = 8) classified as follows: M1—happy, M2—aggressive, M3—calmed status, post-exposure to a stressful situation, compared to the reference time referred to as M0. There were no significant differences (*p* ≥ 0.05) regarding the serotonin values between the two groups of medium and small breed shelter dogs. Following anesthesia, the average mean serotonin values were significantly lower (*p* ≤ 0.003), by 63.85 ng/mL, in stray dogs compared to dogs with owners. No significant differences (*p* ≥ 0.05) were found when comparing the reference time M0 to M1, M2, and M3. The differences decreased significantly (*p* ≤ 0.05), by 89.61 ng/mL, between M1 and M2 and increased significantly (*p* ≤ 0.008), by 112.78 ng/mL, between M2 and M3.

## 1. Introduction

Dogs and humans have coexisted for at least 15,000 years [1], developing close attachment relationships [2]. The development of behavioral issues, which could prompt the owner to surrender the dog to a shelter, can occasionally have a negative impact on this bond [3]. The most frequent behavioral causes of relinquishment include intra- and interspecific aggressiveness, phobias, and separation issues [4]. Many factors can contribute to these issues, including impaired serotoninergic pathway function in the brain, which is defined by serotonin (5-HT) deficit [5,6].

Aggressiveness is defined as a normal behavior of the dog that is part of the ethogram (e.g., a mother that protects her puppies, two males that fight for reproduction, or a dog that defends his own food), but this term is best defined within a given context as appropriate or inappropriate threat or challenge that is ultimately resolved by combat or deference [7]. Canine aggressiveness has been the subject of many studies because it is the most common behavior presented as a problem and the only one responsible for human injury or even death [8,9,10]. Although aggressiveness is a normal behavioral element in a dog pack, it is one of the most frequent reasons why dog owners request the help of veterinarians specializing in behavioral therapy [11,12]. The neurotransmitters that are most frequently studied in animal aggressiveness are dopamine and serotonin. Serotonin is produced from tryptophan and is widely believed to be important in the etiology and treatment of most disorders in dogs.

In the mammal brain, tryptophan is the precursor of serotonin (5-hydroxytryptamine) [13] and plays the role of modulator in neural information processing [14]. Serotonin is considered to be the neurotransmitter that controls several types of behavior: aggressiveness, impulsivity, food selection, stimulation, sexual behavior [15], and reaction to pain. Moreover, serotonin is also involved in the control of emotional manifestation [16].

Because behavior is the consequence of central nervous activity, it is not surprising that differences in neurotransmitters are associated with differences in behavior [17,18,19,20,21]. Furthermore, lower serum 5-HT levels have been related to a lower frequency of sociable behavior toward humans and to one case of repetitive circling in the pen in shelter dogs [22].

There is also evidence supporting the modulatory role of the serotonergic system in behavioral traits in dogs, because behavioral problems, such as aggressiveness or anxiety, have been associated with low serum serotonin levels compared with controls [6,7,23].

In a study performed by Alberghina et al. [22], results demonstrated that serum 5-HT levels are not significantly influenced by sex, age, or environmental conditions. In a previous preliminary study, 5-HT levels had been found to be significantly higher in shelter dogs than in dogs with an owner. This finding was justified by greater social interactions and by the olfactory presence of conspecifics. In another study focused on serotonin evaluation in dogs, Alberghina et al. [24] found that higher levels of 5-HT were seen in the 3–7 years age group, compared to other age groups, but no significant age-related dissimilarities were found. In the case of dogs, their behavior is influenced by genetics, epigenetics, and environmental factors [25]. There is strong evidence that breeds differ in terms of their behavior [26,27,28,29,30].

De Napoli and Dodman [31] observed an improvement in dog behavior by supplementing the dog feed with tryptophan—a serotonin precursor. In another study, Gazzano et al. [32] found that carbohydrate-based morning meals lead to significantly higher TRP/5LNAAs ratios, persistent for at least 6 h. It is possible that the higher bioavailability of TRP, and the consequent increase of 5-HT in the brain, may have positive effects on dog behavioral problems related to a 5-HT deficit [33].

The aim of our study was to assess the variation of serotonin levels in dogs from three experimental variants, under different environmental conditions, in order to improve the knowledge regarding the use of this hormone as a biomarker during adaptation and throughout different emotional statuses.

## 2. Materials and Methods

The study is based on assessing the variation of serotonin values in three experimental variants.

Variant 1. The shelter dogs group divided into two sub-groups of shelter dogs, namely small (4–10 kg) *n* = 6, and medium (10–20 kg) *n* = 4, mixed breed groups.

Behavioral history is based on information provided by caretakers. Aggressive dogs were not included in the study. The aim was to select dogs that could be adopted. The blood samples were collected from the cephalic vein, in the presence of the caretaker, applying the required restraint methods.

The dogs were housed in kennels with access to both indoor and outdoor areas, without exceeding 4 dogs in 6.5 m^2^. As part of general management and routine tasks, they were fed commercial dry food twice daily. None of the dogs underwent a complete behavioral assessment before the trial. Dogs that displayed aggressive behavior towards humans, or showed signs of physical illnesses or injuries that could affect their behavior in response to manipulation were removed from the experiment.

Variant 2. Two groups of dogs from a veterinary clinic, namely, dogs with an owner (*n* = 10) and dogs without owners that were in foster care at the moment of study (*n* = 15), prior to spaying/neutering. The dogs in this experimental variant were brought to the clinic by the owner or by the foster caregiver. Blood samples were collected in the veterinary clinic, while the animals were under anesthesia. The difference between the two groups in Variant 2 was the absence/presence of the owner before the induction of anesthesia.

Variant 3. A group of dogs in different behavioral states (*n* = 8). The dogs belonging to this experimental variant all had an owner. The experiments were conducted on the dog’s territory, in the presence of the owner. The owners completed a behavioral questionnaire, thus providing information related to the behavioral history of the dogs. Each of the eight dogs from this group had a catheter inserted into their cephalic vein, and blood samples were drawn four times to observe changes in serotonin levels.

In order to see how serotonin levels vary based on the emotional state of the dogs, we put them in situations where they were calm, relaxed, or, conversely, stressed. Samples were taken while recording behavioral data. The first blood sample was drawn five minutes after the catheter was fitted. This moment was referred to as the zero moment (M0). At moment 1 (M1) blood was collected after putting the dog in a good mood, which was induced through play with the owner or by giving them their favorite treat. Moment 2 (M2) was the moment when the dog was put in an agitated, stressed (or uncomfortable) situation, such as: putting it in the presence of a stranger or that of another dog, or following rough play with the owner. Depending on their temper, their reactions ranged from irritable to aggressive. Indicators of distress displayed by agitated dogs include horripilation, staring, tensed eyebrows, growling, baring their teeth, and biting attempts. The presence of these signs suggested the perfect timing to draw blood. The final blood-collection moment (M3) was considered the moment when the dog became less distressed and appeared calm.

A period of ten minutes passed between each blood collection moment. During this time, the dog was kept in his owner’s proximity.

In this study, there were equal numbers of male and female dogs, between the ages of 4 and 6 years. The health status of the studied dogs was assessed through physical, clinical, and paraclinical exams. Blood samples were collected only from healthy dogs. From a behavioral point of view, the information was obtained in different ways for each experimental variant. In the variant of shelter dogs, the caregivers provided details related to the behavior of each individual. In Variant 2, the dog owners who came with their pets for the surgical procedure filled in a questionnaire including details about their dog’s behavior. For the dogs without an owner, information was provided by their foster caregivers. For the dogs belonging to Variant 3, each owner completed a behavioral questionnaire.

The blood samples were collected in 6 mL vacutainers without anticoagulant, from the cephalic vein, using a 19G needle. The blood samples were kept at room temperature for two hours. After that, they were centrifuged for five-minutes at 1380 g. The resulting serum was collected using Pasteur pipettes and was placed in Eppendorf tubes, to be frozen at −40 °C. The samples were kept in storage for a maximum of six months before examination. The serum serotonin was determined with the help of a MikroWin 2010 device, Version 5, type Crocodile, producer Titer Tek- Berthold Germany, open system. The utilized kit was a Serotonin ELISA Enzyme Immunoassay for the Quantitative Determination of Serotonin in Serum, Plasma, and Urine from DLD Diagnostika, a company located in Hamburg, Germany. The analyses were performed in the Metabolic Studies Laboratory from the Horea Cernescu Laboratory Complex, part of BUASVM Timişoara. In order to test the significance of the recorded variations, we used the 7 Statistic program and the Mann–Whitney U test, which is a non-parameter test, to evaluate the differences between small groups. The result was considered significant if *p* ≤ 0.05.

This study was approved by the Bioethics Board from Timisoara, no. 57, from 13 July 2021.

## 3. Results

### 3.1. Variant 1

The dynamics of individual values obtained in dogs from the first variant are shown in Table 1 and Figure 1. In the case of shelter dogs, the mean serotonin value was 397.62 ± 167.72 ng/mL in the medium breed group and 343.73 ± ng/mL in the small breed group. The 53.89 ng/mL difference between the two dog groups was insignificant (*p* ≥ 0.39).

### 3.2. Variant 2

In the second variant, shown in Table 2 and Figure 2, dogs without owners had lower individual serotonin values than dogs with owners in 75% of cases.

Dogs without owners did not display aggressive behavior prior to anesthesia. The mean serotonin values were 229.04 ng/mL. In the case of dogs that had an owner, the mean serotonin values were 292.89 ng/mL, the 63.85 ng/mL difference being distinctly significant (*p* ≤ 0.003). The groups showed the same homogeneity, with a CV value close to 17%.

### 3.3. Variant 3

The results of the statistical data processing for the third variant are shown in Table 3 and Figure 3.

M0, represented a reference moment for each individual-the average registered serotonin value was 240.41 ng/mL with a high variation limit of 113.35 ng/mL to 483.02 ng/mL. The variability coefficient was high, 47.8%.

M1, a state of joy—the average serotonin value increases to 264.32 ng/mL and the limit values range from 149.74 ng/mL to 452.42 ng/mL. Although variability is lower compared to the prior moment, it remains high, at 38.88%. We observed a decrease in serotonin levels in three subjects compared to M0 (Dogs 3, 5, and 6), while the other five subjects showed an increase in serotonin levels (Dogs 1, 2, 4, 7, and 8).

M2, aggressiveness—the average serotonin value decreases to 174.71 ng/mL, with values ranging from 111.73 ng/mL to 298.17 ng/mL. The variability coefficient of 34.86% is lower than that recorded for M0 and M1, but it remains high. The aggressiveness thus triggers only a slight homogeneity increase within the group. The dogs react differently, because they are different individuals. All eight dogs present lower serotonin levels than those recorded for M1.

Behavioral display—depending on their environment, the dogs present signs of restlessness, such as horripilation, staring, tensed eyebrows, growling, teeth baring, and biting attempts. Firstly, the owners completed a behavioral questionnaire to provide information related to the behavioral history of each dog. During the experiments, the dogs were filmed and photographed, and the resulted data were subsequently analyzed individually, resulting in the interpretation of the body language of each dog. Thus, the resulted inventory contained the behaviors of each individual.

M3, dogs calmed down by their owners—the average serotonin level increases to 273.27 ng/mL, with values ranging from 170.43 ng/mL to 404.30 ng/mL. The variability coefficient is at its lowest, −28.92%, compared to the other three states; however, the group is still not homogenous. All eight dogs present higher serotonin levels than during M2.

Statistically significant results were found between M2 and M3 when serotonin values increased by 112.78 ng/mL. There was a significant serotonin decrease between M1 and M2 of 89.61 ng/mL, and an insignificant increase of 23.17 ng/mL compared to M3. Even though it was not statistically significant, we found an increase in serotonin level, by 23.92 ng/mL, between M0 and M1, followed by another statistically insignificant increase of 65.69 ng/ ml, compared to M2. In comparison to M3, there was an insignificant increase of 47.08 ng/mL (Table 4).

In terms of typical behavior manifestations, we observed an immediate lack of reaction to stimuli that should induce aggressiveness, exaggerated aggressiveness towards foreigners, timidity, and territorial aggressiveness.

If the dogs were to undergo the adoption preparation process, they would first have to be taken away from the common enclosure and grouped at least according to size and primary behavior manifestation (aggressive/nonaggressive), so as to continue the process of adapting to a human afterwards. Only after this stage could it be established whether any dog under study could adapt to the requirements of becoming part of a future family.

## 4. Discussion

### 4.1. Variant 1

The average serotonin value in medium breed shelter dogs was 397.62 ng/mL, similar to the results obtained by Alberghina et al. [24], who determined average serotonin values of 400 ng/mL. Riggio et al. [34] observed no association between serum TRP and 5-HT levels in their sample of shelter dogs. Because it is widely accepted that TRP and 5-HT concentrations are correlated in the brain, negative findings may be at least partially explained by the role of the blood–brain barrier, regulating the passage of TRP from peripheral to central circulation and that of 5-HT in the opposite direction. On this topic, there is a continuous tendency to try to establish a link between behavior, the peripheral level of serotonin, and tryptophan. Only future studies will be able to give an explanation about the mechanisms underlying the action of serotonin on behavior. Currently, only assumptions can be made.

The results of this pilot study should be interpreted with caution considering the limited number of tested dogs. In order to draw a solid conclusion, further studies are recommended, involving a higher number of animals. A higher serotonin value of 400 ng/mL was determined in four dogs from the medium breed group, and in two of them the determined values were close to 200 ng/mL. In this category, the CV% (coefficient of variability) was 42.18%, indicating a non-homogeneous group, with an unstable group hierarchy, leading to aggressiveness.

The small breed group presented a better homogeneity, a fact mirrored by the CV% value of 15.26%. No individual showed values above 400 ng/mL; however the recorded values, exceeding 300 ng/mL, indicate a mild group instability, represented by insufficiently consolidated relationships (which should induce a state of tranquility). Due to the shelter management, the dogs were not housed according to their sex, age, or health status, and there were constant changes, making it impossible to establish a stable hierarchical structure. When talking about the current shelter, one may mention a continual struggle for hierarchy establishment, because there is neither a social structure originating from a maternal line, as in a pack, nor a family type structure, where the human–animal relationship is more stable.

### 4.2. Variant 2

The lower serotonin level in the group of dogs without owners can be influenced by emotional instability triggered by contact with a new environment prior to anesthesia, and the lack of a stability factor represented by the owner’s presence. The fact that dog number 10 displayed an aggressive behavior towards the accompanying person after undergoing surgery might be explained by the partial metabolization of the anesthetic. No other incidents were recorded. The higher serotonin level recorded in the dogs accompanied by their owner can be related to the fact that they were used to socializing and, thus, were prepared to deal with novel situations. Socially integrated dogs (dogs with owners) displayed a value of 292.89 ng/mL, even during stressful moments, as compared to the non-integrated ones, whose serotonin level tended towards 400 ng/mL, according to Alberghina et al. [24].

The research carried out by Leon et al. [5] revealed that aggressive dogs presented the lowest serotonin levels, 209.6 ng/mL, while for non-aggressive dogs, the recorded value was of 282.5 ng/mL. The lowest serotonin values were determined in dogs that manifested defensive aggressiveness. During the same stress conditions determined by spaying/neutering, the serotonin value was significantly higher in dogs with owners, exceeding that of dogs without an owner by 27.88 %.

We may thus conclude that, generally, the owner’s presence before anesthesia triggered an increase in serotonin levels.

Different results were obtained by Rosado et al. [23], who classified dogs in four categories, depending on the manifested aggressiveness and the recorded serum serotonin levels: dogs manifesting aggressiveness towards family members (SCA), showing a serotonin level of 277.7 ng/mL; defensive aggressiveness towards foreigners (DA), 235.8 ng/mL serotonin level; offensive aggressiveness towards unknown persons (OA), 330.8 ng/mL serotonin level; other forms of aggressiveness, showing a 345.1 ng/mL serotonin level; control group, with a 387.4 ng/mL serotonin level.

### 4.3. Variant 3

At reference moment M0, the average serotonin level was 240.41 ng/mL. The reason for the high variability coefficient, of 47.8%, is that each dog responded differently to handling and blood sampling.

From a behavioral perspective, it was noticed that the dogs did not exhibit indicators of discomfort during M1 when they were happy; their eyes were lively, they were looking at their owners, and their ears were in an upright, but not erect, position.

The results obtained by us during the last experiment point out the fact that during the state of aggressiveness displayed at M2, the average serotonin level was 174.71 ng/mL, significantly lower than the one obtained by Rosado et al. [23].

When the owners calmed their dogs during M3, the average level of serotonin increased, as expected, to 273.27 ng/ml. The dogs that displayed significant variations in their serotonin levels during the four moments were anxious individuals that, due to temporary discomfort, altered their behavior, with a visible impact on their serotonin levels.

The environment, the owner’s personality, and the handling manner of each dog were factors that directly influenced each individual behavior, albeit in a different manner. Some authors have reported behavioral similarities between owners and their dogs when analyzing aggressive behavior [35,36,37].

Amat et al. [6] compared serum serotonin levels in aggressive English Cocker Spaniels with those of aggressive dogs of a variety of other breeds and found that serotonin levels were significantly lower in the Cocker. In another scientific paper, published by Leon M. et al. [5], different methods for determining peripheral serotonin were compared, and correlations were made between the serotonergic system and dog aggressiveness.

Serotonin values observed in aggressive dogs were lower than in non-aggressive dogs, suggesting an inverted correlation between the serotonergic system and aggressiveness. In our study, during experimental Variant 3, which represented the moment of aggressiveness display, a decreasing tendency was observed in serotonin values, compared to the other moments. These observations are in accordance with the studies of other researchers. The serotonergic system and the hypothalamic-pituitary-adrenal axis are considered key players in controlling aggressiveness.

What is a good temperament for a guard dog is not necessarily a good temperament for a pet dog. In fact, most of the commonly used temperament tests rely on the dog’s reaction to foreign people and dogs, and do not measure the reaction of the dogs to challenges posed by the owner [38].

In the research conducted by Karpiński, M. et al. [39], it was found that the plasma noradrenaline, cortisol, and serotonin levels were lower in female dogs than in males. Additionally, the plasma noradrenaline and serotonin levels were higher in the right-pawed dogs than in the left-pawed dogs. The results confirm the assumption that right-pawed dogs adapt to stressful conditions more readily, even if some authors suggest that males are more frequently referred to animal behaviorists due to aggression problems than do females [9,10]

Blood prolactin, serotonin, and oxytocin may represent biomarkers to assess workload and chronic stress-related responses in assistance dogs and eventually improve their selection and training. In addition, using neurohormonal profiling to evaluate their emotional responses to the workload and challenging situations they face during their training may help assess their welfare. This may conduct to an improvement of the educational programs and, hopefully, in the longer term, to establish guidelines about the training of assistance dogs [40]. The recorded serum serotonin levels in the three experimental variants of the present study add data to the current knowledge, suggesting that the obtained values can be used to better identify dogs with an aggressive potential, especially in the case of shelter dogs and dogs with owners, with frequently occurring behavioral problems.

In addition to breed-related variation, environmental input and individual learning have been shown to have a substantial effect on aggressive [41], impulsive [42], and fearful [43] behavior in dogs. Furthermore, environmental factors such as exposure to toxins [44] and enriched versus impoverished living conditions [45] have been linked to changes in the serotonergic system in mammals. Given the variety of environments that pet dogs originate from, it is likely that at least some of these factors will affect either behavior or serum serotonin levels as much or more than any inherent covariation that may exist between them [46]. Previously conducted investigations demonstrated that there are many external and internal contributing factors (e.g., breed, size, circumstance, health condition) that greatly influence canine behavior and, subsequently, their serum serotonin level at a given moment.

## 5. Conclusions

The results of our study demonstrated the presence of statistically significant differences between: (i) the decrease in serotonin levels of dogs without owners compared to dogs that had an owner, as well as between various situations such as passing from the state of joy to the state of aggressiveness; and (ii) increases of serotonin during passage from the state of aggression to the owner-induced state of calmness. However, no correlations were recorded between the (i) average serotonin level of medium breed shelter dogs and small breed dogs, nor between (ii) serotonin values recorded during the states of joy, aggressiveness, and calmness induced by the owner. Likewise, the serotonin values determined by dog size and behavioral state presented a strong individual influence. The obtained results offer useful insights in reducing the difficulties met during the adoption process. However, further studies are still necessary, involving a considerably greater number of animals, in order to clarify the extent to which serotonin could be successfully used as a biomarker during the adoption process and to predict the emotional statuses of these dogs in addition to their behavioral assessments.

## Figures and Tables

**Figure 1 vetsci-09-00523-f001:**
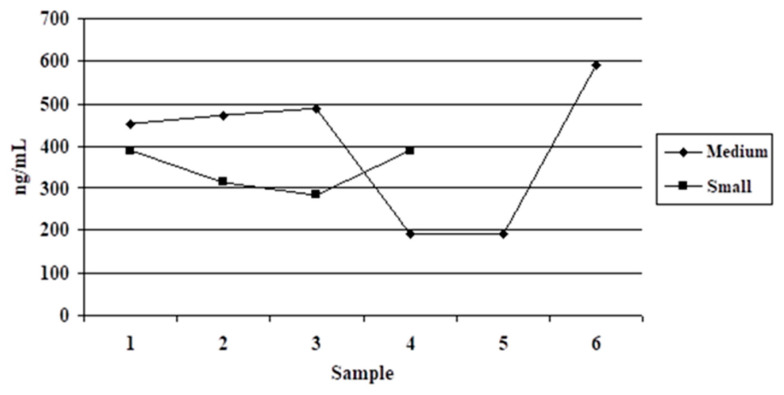
Serotonin dynamics in shelter dogs.

**Figure 2 vetsci-09-00523-f002:**
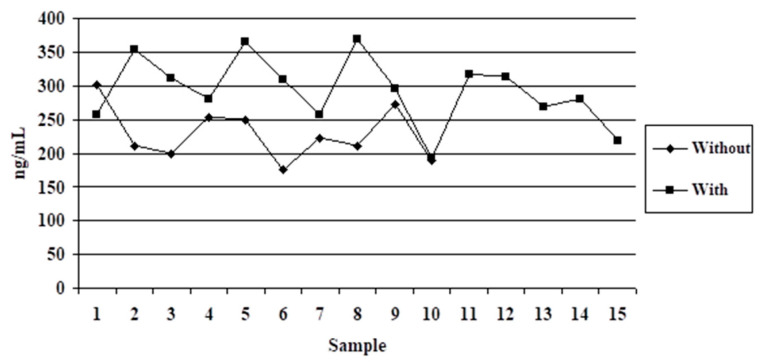
Serotonin dynamics in dogs with or without owner.

**Figure 3 vetsci-09-00523-f003:**
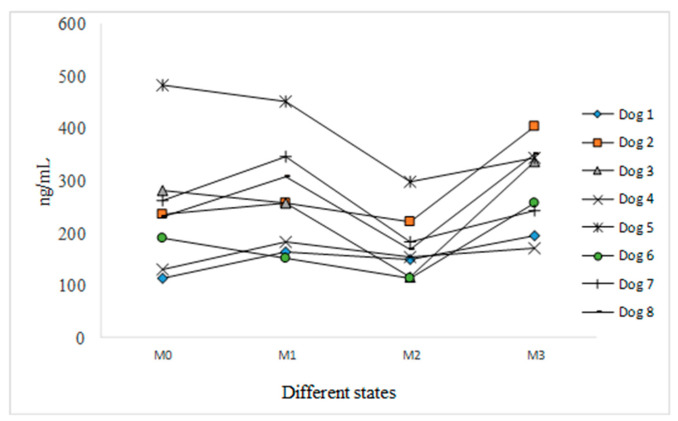
Serotonin dynamics in different states.

**Table 1 vetsci-09-00523-t001:** Serotonin values in dogs depending on their size (ng/mL).

Medium Size (10–20 kg) Group			Small Size (4–10 kg) Group			Difference
*n*	X ± SD	CV%	*n*	X ± SD	CV%	(ng/mL)
6	397.62 ± 167.72	42.18	4	343.73 ± 52.45	15.26	53.89 ns

ns—insignificant (*p* ≥ 0.39). CV—coefficient of variability.

**Table 2 vetsci-09-00523-t002:** Serotonin levels in dogs without or with owner (ng/mL).

Without Owner			With Owner			Difference
*n*	X ± SD	CV %	*n*	X ± SD	CV %	(ng/mL)
10	229.04 ± 40.28	17.59	15	292.89 ± 50.29	17.17	63.85 **

** significant (*p* ≤ 0.003). CV—coefficient of variability.

**Table 3 vetsci-09-00523-t003:** The results of the serotonin value depending on the degree of individual involvement.

States	Results			
	n	X ± SD	CV %	%
M0	8	240.41 ± 114.92	47.80	100
M1 (happy)	8	264.32 ± 102.77	38.88	109.95
M2 (aggressive)	8	174.71 ± 60.91	34.86	72.67
M3 (calmed by owner)	8	287.49 ± 83.15	28.92	119.59

**Table 4 vetsci-09-00523-t004:** The statistical significance of differences between various degrees of individual involvement.

Different States	M0	M1	M2	M3
M0	-	+23.92 ns	−65.69 ns	+47.08 ns
M1	+23.92 ns	-	−89.61 *	+23.17 ns
M2	−65.69 ns	−89.61 *	-	+112.78 **
M3	+47.08 ns	+23.17 ns	+112.78 **	-

ns—insignificant (*p* ≥ 0.05); * significant (*p* ≤ 0.05); ** significant (*p* ≤ 0.008). M0—reference moment; M1—happy; M2—aggressive; M3—calmed by owner.

## Data Availability

The data presented in this study are available in the manuscript.

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
