# Peer review of "The Variation of Serotonin Values in Dogs in Different Environmental Conditions"

_vetsci, 2022, doi:10.3390/vetsci9100523_

Round 1

Reviewer 1 Report

This manuscript describes a study on the evolution of serotonin values in dogs and therefore fits in the scope of the journal. The manuscript, however, lacks detailed information, clarity in writing and results, and is therefore very hard to read and understand. Improvement of the English also might help. The introduction needs considerable improvement, the material and methods need a lot more details, as also the results and discussion.

The effect of serotonin levels on behaviour, such as aggressive behaviour are inconsistent decribed in the introduction, results and discussion.

Author Response

Reviewer #1

This manuscript describes a study on the evolution of serotonin values in dogs and therefore fits in the scope of the journal. The manuscript, however, lacks detailed information, clarity in writing and results, and is therefore very hard to read and understand. Improvement of the English also might help. The introduction needs considerable improvement, the material and methods need a lot more details, as also the results and discussion.

The effect of serotonin levels on behaviour, such as aggressive behaviour are inconsistent decribed in the introduction, results and discussion.

We sincerely thank you for taking the time to review this manuscript, and for your close attention in highlighting the mentioned concerns. We highly appreciate your overall positive feed-back regarding the quality of the manuscript.

During the revision process the authors improved the manuscript quality taking into consideration the comments of other two reviewers. The revised version of the manuscript contains additional information in each of the above-mentioned sections of the manuscript (e.g. introduction, materials and methods, results, discussion, conclusions), offering a clearer and more understandable scenario for the reader. Also, the English content of the manuscript has been revised resulting in significant improvements. The research team hopes that all the raised concerns of the respected reviewers have been successfully addressed. However, we are ready to further revise/modify the article if any shortcomings remain.

Please see the revised version!

Thank you again!

Reviewer 2 Report

This study was focused on the neurotransmitter serotonin level changes among dog sizes and behavioral displays. And, the serotonin levels were measured not in the dog's brain but in the peripheral blood. The correlation between brain serotonin level and peripheral serotonin level is still being demonstrated. Even the origin of peripheral serotonin was still under discussion. However, it was interesting that the blood serotonin levels were different among shelter dogs' emotional states. So, this study is probably acceptable.  I look forward to seeing the mechanism of peripheral serotonin changes in several dogs' emotional statuses.

Author Response

Reviewer #2

This study was focused on the neurotransmitter serotonin level changes among dog sizes and behavioral displays. And, the serotonin levels were measured not in the dog's brain but in the peripheral blood. The correlation between brain serotonin level and peripheral serotonin level is still being demonstrated. Even the origin of peripheral serotonin was still under discussion. However, it was interesting that the blood serotonin levels were different among shelter dogs' emotional states. So, this study is probably acceptable. I look forward to seeing the mechanism of peripheral serotonin changes in several dogs' emotional statuses.

We sincerely thank you for taking the time to review this manuscript, and for your close attention in highlighting the mentioned concerns. We highly appreciate your overall positive feed-back regarding the quality of the manuscript.

During the revision process, the authors improved the manuscript quality taking into consideration the comments of other two reviewers. The revised version of the manuscript contains additional information, offering a clearer and more understandable scenario for the reader. As the reviewer mentioned, in his last comment, the following sentence was inserted at the end of the conclusion section: “However, further studies are still necessary, involving a considerably greater number of animals, in order to clarify the extent to which serotonin could be successfully used as a biomarker during the adoption and emotional statuses of these dogs, in addition to their behavioral assessments.”

Please see the revised version!

Thank you again!

Reviewer 3 Report

Title: The evolution of serotonin values in different behavioral states in dogs 

Recommendation: Moderate Revision.

Overview and general recommendation:

This manuscript investigates the serotonin levels in three different experimental condition: between small and medium shelter dogs; in dogs with and without owners after administration of pre-spaying/ neutering anesthesia; in dogs in different behavioral states before and after exposition to a stressful situation.

The paper covers a very interesting topic in the field of dog behavioural medicine, which is suitable for this journal, and refers to most relevant literature in the area but the three experimental setting are very different one with each other. The three settings do not appear to be connected to each other making the paper a little bit confused. Moreover, the discussions are not really focused on the results and the conclusions are not adequate. It could be useful to discuss the results of each experimental setting in a different paragraph trying to use a more cautious approach in the discussion and conclusion section.

Specific comments

Title

The title is focalized just on your third experimental setting: it could be better using a title more generic including all your experiments. Moreover, the term “evolution” is not really appropriate, it is better changing it with “variation”.

Keywords

-Please modify “neurotransmitters level”: is not really your topic

 Please add “serotonin”.

Introduction

-Line 37: please delete “genetics” 

-Lines 37-44: Aggressiveness is a normal behavior of the dog that is part of the ethogram, see a mother that protects her puppies, two males that fight for reproduction, a dog that defends his own food, ecc. Aggression is best defined within a given context as appropriate or inappropriate threat or challenge that is ultimately resolved by combat or deference. So, it is better using the terms pathological aggression speaking of behavioral disorders (Overall K. L. 2013. Manual of Clinical Behavioural Medicine for dogs and cats. Elsevier). Please consider this aspect in your introduction. 

-Lines 67-48: Can you give a brief explanation?

-Lines 69-76: This part does not appear well amalgamated with the rest...can you try to correlate it better?

-Line 77: Please replace “evolution” with “variation”

-Lines 77-84: Please try to review the aims of your work and try not to draw too much of a spurious conclusion from it. It could be better discussing the topic regarding the advantage of knowing the serotonin level and variation in dog adoption in the discussion section rather than in this part of the text.

Materials and Methods

-Line 86: Please replace “evolution” with “variation” 

-Lines 88-93: Please provide more details regarding the environment and management of these dogs (How were the dog pens? How many dogs were inside? How was da daily routine? etc.)

-Lines 94-98: For me is not clear this experimental setting…could you explain better? Are the dogs all owned or some of them were shelter dogs? 

Line 97: please replace “from variant 2” with “in variant 2” 

-Lines 99-116: please add more information regarding the dogs (es. Where the dogs come from? Where was the experiment conducted? etc.

-Lines 117-119: Are these sentences referred just to the variant 3 or to all the variants? 

Results

It might be useful to write both the results and the discussions by making sub-paragraphs related to the single variants of your work; this could help in reading the paper making things clearer.

-Lines 149-156: could you explain better this part? Part of these sentences should be moved in discussions section and justified with data related to the management of dogs. 

-Line 176: please replace “Mo” with “M0” 

-Lines 189-191: Please describe better how you collected and analysed the behavioural data.

-Lines 201-211: it could be better specifying just the significant results and describe the other using a sentence like “Even if not statistically significant, we found ……”, not providing for them the P value.

-Lines 212-214: How do you know they’re anxious dogs? Is that a diagnosis or your assumption? According to me, you could move this part in the discussion section.

Discussion

It might be useful to write both the results and the discussions by making sub-paragraphs related to the single variants of your work.

-Lines 219-225: Could you explain better these concepts? 

-Lines 226-230: It would be better to use more caution in the interpretation of the results and to indicate them as results of a pilot study that should be deepened and improved increasing the sample 

-Lines 231-236: you could discuss this part together with that mentioned in the results and move it after the discussion of all variants, to create a union and an amalgam in the discussion of your results.

-Lines 237-245: This part is not clear in the M&M and Results section: you need to explain better how you collected and analysed these data. Moreover, you need to use a more cautious approach because you can just suppose it.

-Lines 278-279: Could you explain how?

-Lines 280-288: please try to focus and link better these references to your results.

-Line 289: “attack dog” …do you mean “guard dog”? 

-Lines 293-306: Could you explain better your thought regarding these references? You need to discuss them in relation to your results.

-Lines 315-317: please rephrase this sentence. This assumption is based on previous works and previous literatures not only on your work. Moreover, please change the word “evolutionary”, it is not the right term.

Conclusion

You need to rewrite the conclusion: you are just reassumed your results. Please delete all numbers and P value in this section. Try to reassume your results giving a quick explanation of it in relation to the present literature and trying to say which is the practical aspect of what you found. You can discuss also the limits of your study and try to find possible solution in future.

-Lines 328-329: You can use this sentence and expand it.

-Line 329: Please use more cautions in this sentence and try to explain why your work will help to reduce the difficulties in adopting dogs.

Figures:

-Figure 1 e 2: please explain better what you mean with “dynamics” 

Author Response

Reviewer #3

Title: The evolution of serotonin values in different behavioral states in dogs

Recommendation: Moderate Revision.

Overview and general recommendation:

This manuscript investigates the serotonin levels in three different experimental condition: between small and medium shelter dogs; in dogs with and without owners after administration of pre-spaying/ neutering anesthesia; in dogs in different behavioral states before and after exposition to a stressful situation.

The paper covers a very interesting topic in the field of dog behavioural medicine, which is suitable for this journal, and refers to most relevant literature in the area but the three experimental setting are very different one with each other. The three settings do not appear to be connected to each other making the paper a little bit confused. Moreover, the discussions are not really focused on the results and the conclusions are not adequate. It could be useful to discuss the results of each experimental setting in a different paragraph trying to use a more cautious approach in the discussion and conclusion section.

We sincerely thank you for taking the time to review this manuscript, and for your close attention in highlighting the mentioned concerns. We highly appreciate your overall positive feed-back regarding the quality of the manuscript.

Please find attached our responses to your comments below, in a point-by point manner:

Specific comments

*Title

The title is focalized just on your third experimental setting: it could be better using a title more generic including all your experiments. Moreover, the term “evolution” is not really appropriate, it is better changing it with “variation”.

According to the reviewer suggestion, the manuscript title was changed resulting in “The variation of serotonin values in dogs under different environmental conditions”

The word “evolution” was replaced with “variation”, as recommended.

*Keywords

*Please modify “neurotransmitters level”: is not really your topic. Please add “serotonin”

The terms were modified, as recommended!

*Introduction

*Line 37: please delete “genetics”

The word “genetics” was deleted!

*Lines 37-44: Aggressiveness is a normal behavior of the dog that is part of the ethogram, see a mother that protects her puppies, two males that fight for reproduction, a dog that defends his own food, ecc. Aggression is best defined within a given context as appropriate or inappropriate threat or challenge that is ultimately resolved by combat or deference. So, it is better using the terms pathological aggression speaking of behavioral disorders (Overall K. L. 2013. Manual of Clinical Behavioural Medicine for dogs and cats. Elsevier). Please consider this aspect in your introduction.

The authors followed the reviewer recommendation, and the following sentence was inserted in the Introduction chapter: “Aggressiveness is defined as a normal behavior of the dog that is part of the ethogram (e.g. a mother that protects her puppies, two males that fight for reproduction or a dog that defends his own food), but this term is best defined within a given context as appropriate or inappropriate threat or challenge that is ultimately resolved by combat or deference” Please see the lines 53-57 of the revised version.

*Lines 69-76: This part does not appear well amalgamated with the rest...can you try to correlate it better?

Lines 70-72 (from the initial submission) were deleted, thus the remaining text in the revised version is in the same context.

*Line 77: Please replace “evolution” with “variation”

“Evolution” was replaced with “variation”!

*Lines 77-84: Please try to review the aims of your work and try not to draw too much of a spurious conclusion from it. It could be better discussing the topic regarding the advantage of knowing the serotonin level and variation in dog adoption in the discussion section rather than in this part of the text.

According to the reviewer’s suggestion, the study aim was redefined as follows: “The aim of our study was to assess the variation of serotonin levels in dogs from three experimental variants, under different environmental conditions, in order to improve the knowledge regarding the use of this hormone as a biomarker during the adaptation and throughout different emotional statuses”. Materials and Methods

*Line 86: Please replace “evolution” with “variation”

“Evolution” was replaced with “variation”!

*Lines 88-93: Please provide more details regarding the environment and management of these dogs (How were the dog pens? How many dogs were inside? How was da daily routine? etc.)

As answer to the reviewer question the following information was inserted in the revised version, lines 109-114: “The dogs were housed in kennels with access to both indoor and outdoor areas, without more exceeding 4 dogs in 6.5 m2. As part of general management and routine tasks, they were fed commercial dry food twice daily. None of the dogs underwent a complete behavioral assessment before the trial. Dogs that displayed aggressive behavior towards humans, or showed signs of physical illnesses or injuries, that could affect their behavior in response to manipulation, were both removed from the experiment”.

*Lines 94-98: For me is not clear this experimental setting…could you explain better? Are the dogs all owned or some of them were shelter dogs?

The dogs in variant 2 were divided as follows: 10 dogs had owners and were brought to the clinic for neutering, 15 dogs did not have an owner, they were brought in by the foster caretaker, also for neutering, in preparation for the adoption process. A clearer explanation was inserted in the revised version (lines 115-117): “Two groups of dogs from a veterinary clinic namely, dogs, with an owner (n = 10) and dogs without owners that were in foster care at the moment of study (n = 15), prior to spaying/neutering.

*Line 97: please replace “from variant 2” with “in variant 2” 

“From variant 2” was replaced with “in variant 2”.

*Lines 99-116: please add more information regarding the dogs (es. Where the dogs come from? Where was the experiment conducted? etc.

The authors inserted the following information in the revised version (lines 122-125): „The dogs belonging to this experimental variant had an owner, the experiments were conducted on the dog's territory, in the presence of the owner. The owners completed a behavioral questionnaire, thus providing information related to the behavioral history of the dogs.

*Lines 117-119: Are these sentences referred just to the variant 3 or to all the variants?

The sentence refers to all dogs involved in this study.

Results

*It might be useful to write both the results and the discussions by making sub-paragraphs related to the single variants of your work; this could help in reading the paper making things clearer.

Changes done, as recommended!

*Lines 149-156: could you explain better this part? Part of these sentences should be moved in discussions section and justified with data related to the management of dogs.

According to the reviewer’s requirement, the mentioned text from the results section of the originally submitted version was moved to the discussion section, and the following explanation was added (lines 271-274): “Due to the shelter management, the dogs were not housed according to their sex, age, or health status, and there were constant changes, making it impossible to establish a stable hierarchical structure.”

*Line 176: please replace “Mo” with “M0” 

“Mo” was replaced with “M0” according to the reviewer’s suggestion.

*Lines 189-191: Please describe better how you collected and analysed the behavioural data.

“Firstly, the owners completed a behavioral questionnaire to provide information related to the behavioral history of each dog. During the experiments, the dogs were filmed and photographed, and the resulted data were subsequently analyzed individually, resulting in the interpretation of the body language of each dog. Thus, the resulted inventory contained the behaviors of each individual” (please see the lines 213-217, in the revised version).

*Lines 201-211: it could be better specifying just the significant results and describe the other using a sentence like “Even if not statistically significant, we found ……”, not providing for them the P value.

We changed the content without providing for them the P value resulting in (please see the lines 228-234 of the revised version). Statistically significant results were found between M2 and M3 when serotonin values increased by 112.78 ng/ml. There was a significant serotonin decrease between M1 and M2 of 89.61 ng/ml and an insignificant increase, of 23.17 ng/ml, when compared to M3. Even though it was not statistically significant, we found an increase in serotonin level, by 23.92 ng/ml, between M0 and M1 followed by another statistically  insignifi-cant increase of 65.69 ng/ ml, compared to M2. By comparison to M3, there was an in-significant increase of 47.08 ng/ml.”.

*Lines 212-214: How do you know they’re anxious dogs? Is that a diagnosis or your assumption? According to me, you could move this part in the discussion section.

The dogs were not diagnosed with anxiety, we classified them as such, based on the behavior displayed during the study period. As you recommended, the sentence from lines 212-214 was moved to the discussion section (Please see the lines 316-319).

Discussion

*It might be useful to write both the results and the discussions by making sub-paragraphs related to the single variants of your work.

Changes done, as recommended!

*Lines 219-225: Could you explain better these concepts?

The word “ours” was deleted, it was a typo, so we hope that the phrase, now, has the right meaning. On this topic, there is a continuous tendency to try to establish a link between behavior, the peripheral level of serotonin and tryptophan. Only future studies will be able to give an explanation about the mechanisms underlying the action of serotonin on behavior. Currently, only assumptions can be made” (please see the lines 256-260).

*Lines 226-230: It would be better to use more caution in the interpretation of the results and to indicate them as results of a pilot study that should be deepened and improved increasing the sample

The phrase from lines 226-230 (originally submitted version) was slightly modified and we think that now it has more meaning: “The results of this pilot study should be interpreted with caution considering the limited number of tested dogs. In order to draw a solid conclusion, further studies are recommended involving a higher number of animals”.

*Lines 231-236: you could discuss this part together with that mentioned in the results and move it after the discussion of all variants, to create a union and an amalgam in the discussion of your results.

The paragraph between lines 231-236 was moved according to the reviewer’s recommendations (please see the lines 242-237 of the revised version).

*Lines 237-245: This part is not clear in the M&M and Results section: you need to explain better how you collected and analysed these data. Moreover, you need to use a more cautious approach because you can just suppose it.

The following sentence was inserted in the M&M section (lines 117-118): “The dogs in this experimental variant were brought to the clinic by the owner or by the foster caregiver.”

Changed in the revised version into (lines 279-286): “The lower serotonin level in the group of dogs without owners can be influenced by the emotional instability triggered by the contact with a new environment prior to anesthesia, and the lack of a stability factor represented by the owner’s presence. The fact that dog number 10 displayed an aggressive behavior towards the accompanying person after undergoing surgery might be explained by the partial metabolization of the anesthetic. No other incidents were recorded. The higher serotonin level recorded in the dogs accompanied by their owner can be related to the fact that they were used to socializing and thus, were prepared to dealing with novel situations”.

*Lines 278-279: Could you explain how?

Over time, dogs have evolved to adapt to the social demands of humans. The skills that dogs have acquired have been demonstrated in specialized literature, specifically, the ability to interact as easily as possible with the human species. Dogs, living together with people, form very close bonds with their owners, thus, it can be explained that, sometimes, if the owner is aggressive, the dog can also adopt the same behavior.

*Lines 280-288: please try to focus and link better these references to your results.

Thank you for your valuable comments and suggestions, we added the following sentence (lines 332-335): “In our study, during experimental variant 3, which represented the moment of aggressiveness display, a decreasing tendency was observed in serotonin values, compared to the other moments. These observations are in accordance with the studies of other researchers”.

 *Line 289: “attack dog” …do you mean “guard dog”?

Yes, we meant to say guard dog, and changed it in the text!

*Lines 293-306: Could you explain better your thought regarding these references? You need to discuss them in relation to your results.

According to your suggestions, the authors tried to make a connection between the references and the present study, as follows (lines 354-358): “The recorded serum serotonin levels in the three experimental variants of the present study, add data to the current knowledge, suggesting that the obtained values can be used to better identify dogs with an aggressive potential, especially in case of shelter dogs and dogs with owners, with frequently occurring behavioral problems.

*Lines 315-317: please rephrase this sentence. This assumption is based on previous works and previous literatures not only on your work. Moreover, please change the word “evolutionary”, it is not the right term.

Following the reviewer recommendation, the sentence was rephrased resulting in (lines 366-369): “Previously conducted investigations demonstrated that there are many external and internal contributing factors (e.g. breed, size, circumstance, health condition) which greatly influence canine behavior and, subsequently, their serum serotonin level at a given moment.”

Conclusion

*You need to rewrite the conclusion: you are just reassumed your results. Please delete all numbers and P value in this section. Try to reassume your results giving a quick explanation of it in relation to the present literature and trying to say which is the practical aspect of what you found. You can discuss also the limits of your study and try to find possible solution in future.

The authors completely rewrote the conclusion section according to the reviewer’s suggestion. Please see the lines 371-384 of the revised version.

The results of the present study demonstrated the presence of statistically significant differences between the: (i) decrease in serotonin levels of dogs without owners compared to dogs that had an owner, as well as between various situations such as passing from the state of joy to the state of aggressiveness; and (ii) increases of serotonin during passage from the state of aggression to the owner-induced state of calmness. However, no correlations were recorded between the (i) average serotonin level of medium breed shelter dogs and small breed dogs, nor between (ii) the serotonin values recorded during the states of joy, aggressiveness and calmness induced by the owner. Likewise, the serotonin values determined by dog size and behavioral state presented a strong individual influence. The obtained results offer useful insights in reducing the difficulties met during the adoption process. However, further studies are still necessary, involving a considerably greater number of animals, in order to clarify the extent to which serotonin could be successfully used as a biomarker during the adoption process and to predict the emotional statuses of these dogs in addition to their behavioral assessments.

*Lines 328-329: You can use this sentence and expand it.

This concern was previously answered.

*Line 329: Please use more cautions in this sentence and try to explain why your work will help to reduce the difficulties in adopting dogs.

This concern was previously answered.

Figures:

*Figure 1 e 2: please explain better what you mean with “dynamics”

We used the term "dynamic" to imply that the serotonin value is not constant and can vary depending on a range of conditions.

Thank you again for your efforts and we hope you find our modifications and explanations satisfying!

Round 2

Reviewer 3 Report

Thanks for your work: the authors did a very good job of responding to my requests and comments.